# Brassinosteroids in Micronutrient Homeostasis: Mechanisms and Implications for Plant Nutrition and Stress Resilience

**DOI:** 10.3390/plants14040598

**Published:** 2025-02-17

**Authors:** Laiba Usmani, Adiba Shakil, Iram Khan, Tanzila Alvi, Surjit Singh, Debatosh Das

**Affiliations:** 1School of Health Sciences and Translational Research, Department of Biotechnology, Sister Nivedita University, Kolkata 700156, West Bengal, India; 2Natural Products Utilization Research, U.S. Department of Agriculture, Agricultural Research Service, Oxford, MS 38677, USA

**Keywords:** brassinosteroids, signaling, stress, micronutrients, plant nutrition

## Abstract

Brassinosteroids (BRs) are crucial plant hormones that play a significant role in regulating various physiological processes, including micronutrient homeostasis. This review delves into the complex roles of BRs in the uptake, distribution, and utilization of essential micronutrients such as iron (Fe), zinc (Zn), manganese (Mn), copper (Cu), and boron (B). BRs influence the expression of key transporter genes responsible for the absorption and internal distribution of these micronutrients. For iron, BRs enhance the expression of genes related to iron reduction and transport, improve root architecture, and strengthen stress tolerance mechanisms. Regarding zinc, BRs regulate the expression of zinc transporters and support root development, thereby optimizing zinc uptake. Manganese homeostasis is managed through the BR-mediated regulation of manganese transporter genes and chlorophyll production, essential for photosynthesis. For copper, BRs influence the expression of copper transporters and maintain copper-dependent enzyme activities crucial for metabolic functions. Finally, BRs contribute to boron homeostasis by regulating its metabolism, which is vital for cell wall integrity and overall plant development. This review synthesizes recent findings on the mechanistic pathways through which BRs affect micronutrient homeostasis and discusses their implications for enhancing plant nutrition and stress resilience. Understanding these interactions offers valuable insights into strategies for improving micronutrient efficiency in crops, which is essential for sustainable agriculture. This comprehensive analysis highlights the significance of BRs in micronutrient management and provides a framework for future research aimed at optimizing nutrient use and boosting plant productivity.

## 1. Introduction

Plant development and growth are complex and dynamic processes that incorporate cell division, expansion, and differentiation. They allow plants to adjust to their environmental elements, reproduce, and sustain ecosystems. These processes are regulated by a range of internal elements, including genetic programming and hormonal control, as well as extrinsic ones like light, water, temperature, and soil nutrients [1]. A balanced supply of key nutrients is critical for a plant’s ideal development and growth. The availability of nutrients is a determining factor in the growth, development, and productivity of plants. Macronutrients are the type of nutrients that plants need in large quantities and have been the subject of exploration for most of the research on plant development and growth. On the contrary, micronutrients are required in trace quantities by the plant; however, this quantity does not negate the significant role played by them in regulating the growth of plants [1]. Essential micronutrients, which include iron (Fe), copper (Cu), zinc (Zn), boron (B), and manganese (Mn), have been known to play a crucial role in a range of physiological processes of plants that include but are not limited to cellular respiration, photosynthesis, enzymatic activity, and structural integrity [2]. For instance, iron (Fe) plays a significant role in chlorophyll synthesis and respiration, whereas copper (Cu) is needed for electron transport in photosystem I, the water-splitting reaction of photosynthesis is facilitated by manganese (Mn), and boron (B) is required in membrane integrity and cell wall formation [1]. In addition to their involvement in normal plant growth and development, micronutrients were also found to be essential to the innate immunity of plants and their tolerance against stressors via being engaged in the metabolic processes that regulate the perception of plants to stressors and plant response [3]. Nonetheless, a common agricultural problem that emerges globally is micronutrient deficiencies in soils that are caused by improper fertilization, poor soil conditions, and environmental stressors such as salinity and drought, which eventually lead to poor crop quality, reduced yield, and stunted plant growth.

On the contrary, it is also essential to maintain a balance of the micronutrient concentration in plants as it could lead to micronutrient toxicity, which could further endanger the plant growth and development through reduced growth, oxidative stress, leaf tip necrosis, and more [3]. In this regard, micronutrient homeostasis emerges as a crucial factor that offers optimal availability and proper functioning while preventing toxicity due to overaccumulation by tightly regulating micronutrient uptake, transport, storage, and utilization [4]. Hormonal regulation, specific transporter proteins, and signaling molecules are some of the methods employed by plants to maintain optimum micronutrient levels. While extensive studies on traditional plant growth hormones like gibberellins, auxins, and cytokinins have been conducted in terms of nutrient uptake and plant growth, their scope remains limited in exploring the role played by these hormones in micronutrient homeostasis. Even though many studies have explored the crosstalk between micronutrients and phytohormones, not many have reflected the role played by these phytohormones in micronutrient homeostasis beyond being crucial in managing abiotic and biotic plant stresses. This has caused researchers to investigate plant hormones past the traditional ones, such as brassinosteroids (BRs), that have shown promising roles in plant stress resilience and micronutrient homeostasis.

First found during the 1970s, brassinosteroids are a family of plant hormones known for their ability to stimulate cell division and elongation [5,6]. Since then, they have been more engaged in a range of plant activities, including development, growth, and stress responses [7,8]. The significance of BRs as regulators of plant nutrition, especially in micronutrient homeostasis, is turning out to be well recognized. They are vital for the plant’s ability to effectively deal with micronutrients since they are known to control root and shoot development, enhance photosynthetic productivity, and influence stress resistance [9,10]. The capability of BRs in controlling the expression of transported genes, which are responsible for nutrient absorption and distribution, has been uncovered in recent research, and this assists plants with maintaining an optimal internal balance of micronutrients [10,11]. Growing research on BRs shows that they are fundamental for overall plant health as well as assuming a key role in nutrient efficiency management, which is an essential area for enhancing crop yield and sustainability in contemporary agriculture. This review aims to synthesize the current research on the regulation of micronutrient homeostasis by brassinosteroids, with a particular emphasis on the mechanistic pathways impacting micronutrient uptake, distribution, and utilization.

Other elements, such as selenium (Se) and silicon (Si), are considered beneficial immunomodulators involved in stress tolerance and plant defense mechanisms, but most (not all) are classified as non-essential or conditional micronutrients according to a plant species and environmental context. Additionally, the roles of BRs in the crosstalk with Se and Si are far less known compared to their established functions in regulating Fe, Zn, Cu, Mn and B, which have already been reviewed. This review, centered on these five micronutrients, aims to provide a targeted and in-depth analysis of their interactions with BRs, supported by robust experimental evidence.

This review aims to synthesize current research on the role of BRs in regulating micronutrient homeostasis, with a focus on Fe, Zn, Cu, Mn, and B. By exploring the mechanistic pathways through which BRs influence nutrient uptake, distribution, and utilization, this review provides valuable insights into their potential applications in enhancing plant nutrition and resilience to stress. Understanding these interactions is crucial for developing strategies to optimize nutrient efficiency and improve agricultural productivity sustainably.

## 2. Brassinosteroids: Origin and Importance in Plant Growth

Brassinosteroids (BRs) were found in the 1970s, when researchers from the USDA and Nagoya University, Japan, were pursuing plant growth-promoting compounds. The important finding was isolating brassins from Brassica napus pollen, which induced cell elongation and division, allowing them to discover an active component of brassinolide, a steroidal compound. Brassinosteroids were later identified as important plant hormones controlling many aspects of plant development [12].

Brassinosteroids are polyhydroxylated sterol derivatives, structurally related to animal steroid hormones. The most studied brassinosteroid is brassinolide. Its structure represents a steroidal lactone with a cholestane skeleton (5α-cholestan). The compound contains the following key features (Table 1):

Brassinosteroids (BRs) are plant steroid hormones influencing key physiological processes crucial to plant growth and stress resistance. These hormones play a vital part in sparking seed germination, lengthening stems, growing roots, and shaping overall plant structure. They boost the work of antioxidant enzymes, which build up the plant’s defense against various environmental stresses such as lack of water, too much salt, and extreme heat or cold. Moreover, BRs adjust plant immune responses, helping protect against many harmful organisms. By improving photosynthesis and increasing crop output, BRs show their worth in farming. Because they do not harm the environment and are not toxic, BRs are a great choice for eco-friendly farming methods. As of now, more than 70 epibrassinolides have been identified, which indicates the diversity of brassinosteroids in plant physiology. Among the 70 identified, the only commercially available brassinosteroids are 28-homobrassinolide and 24-epibrassinolide, broadly utilized to improve plant growth, crop yield, and stress tolerance [13] (See Figure 1).

### Signaling Pathways in Plant Growth and Development

Brassinosteroids, or BRs, are important plant hormones that mediate basic growth and developmental changes, such as cell elongation and differentiation in the vasculature, reproductive development, and stress responses. The plasma membrane-initiated signaling event involves receptor kinases and cascades of phosphorylation events and ultimately leads to the nucleus where gene expression is modified. A crucial role is played by brassinosteroid (BR) signaling in the uptake, distribution, and utilization of essential micronutrients such as copper (Cu), zinc (Zn), iron (Fe), boron (B), and more. In the presence of BRs (BR+ve), transphosphorylation is triggered when BRs bind to Brassinosteroid-Insensitive 1 (BRI1) and its co-receptor BRI1-Associated Kinase 1 (BAK1), which eventually leads to the downstream activation of BR-Signaling Kinases (BSKs) and Constitutive Differential Growth 1 (CDG1). This leads to the activation of Brassinazole-Resistant 1 (BZR1) and Brassinosteroid-Enhanced Expression 1 (BES1) transcription factors. These transcription factors help in regulating Zinc-Regulated Transporters/Iron-Regulated Transporters (ZIPs) for the transport of Zn, Iron-Regulated Transporter 1 (IRT1) for the uptake of Fe, and Heavy Metal ATPases (HMAs) for Cu distribution. On the contrary, the absence of BRs (BR-ve) renders BRI1 inactive because of the interaction of BRI1 with BRI1-Kinase Inhibitor 1 (BKI1). Also, Glycogen Synthase Kinase 3 (BIN2) prevents the nuclear localization and transcriptional activation of BZR1 and BES1 by constitutively phosphorylating them. Thus, BR signaling plays a vital role in the uptake and homeostasis of micronutrients in plants (Figure 2) [14,15].

## 3. Implications of Brassinosteroids

Brassinosteroids (BRs) constitute a class of phytohormones known to be present in all land plants. BRs play an important role in various plant physiological and developmental processes. The studies on BRs continue to grow, and they are now showing potential for use in areas such as nutritionally improved plants, stress resilience, and agricultural growth. We critically discuss these implications in further detail using the studies cited.

### 3.1. Plant Nutrition

Brassinosteroids were found to significantly influence nutrient uptake and metabolism, both important parameters for the growth and productivity of plants. For example, BRs stimulate photosynthesis and nitrogen metabolism, especially when the plants are under stress conditions, such as chilling stress. In pepper seedlings, for example, treatment with BR enhanced the efficiency of nutrient assimilation and conversion energy, thus boosting the general growth of plants and photosynthesis activity [17].

BRs enhance the root uptake of major nutrients such as nitrogen and phosphorus by promoting nutrient transport across various organs of plants through modifications in root architecture and the alteration of root growth [18]. This effect in addition to improving plant stress resilience points to the significance of BRs in enhancing the nutritional value of plants.

### 3.2. Stress Resilience

Brassinosteroids have gained much attention due to their role in stress mitigation. They enhance the tolerance capacity of plants toward both abiotic and biotic stresses. For example, under drought conditions, BRs improve water usage efficiency and antioxidant systems of apple varieties to ensure the survival of these plants during drought [19]. Similar tolerance mechanisms to salt stress have also been reported with BRs in *Eucalyptus urophylla*, an improvement that records changes in antioxidant metabolism, the anatomy of the leaves, and total plant homeostasis [20].

BRs also interact with other phytohormones to control plant responses to stress. This interaction of BR and hormones such as abscisic acid, salicylic acid, and ethylene ensures the adaptation of plants to different types of environmental stress [21]. The interaction of BRs with these hormones is very vital for dealing with temperature extremes; BRs counteract the heat stress damage in cereal crops [22].

BR functional analogs have also been prepared to increase drought tolerance. In soybean, such analogs highly enhanced tolerance to drought conditions; hence, yield losses were much reduced under dry conditions [23]. One of the promising aspects of using BR in agriculture is conferring resilience to stresses.

### 3.3. Sustainable Agriculture

Brassinosteroids are currently an intensely pursued tool for the next Green Revolution in maximizing crop yield and quality. Not only does the manipulation of the BR signaling pathway increase plant biomass but it also increases reproductive output, an important component in the improvement of agricultural productivity [24]. This will be specifically pertinent under a global climatic change scenario where stress-adapted biologically versatile crops are pivotal for food security.

Besides promoting growth under normal conditions, BRs protect crops against stress-induced decreases in yield. For instance, the use of BRs enhances the overall physiological condition of horticultural crops by increasing their resistance to abiotic stresses, thereby making them grow even in unfavorable environmental conditions [25]. BRs have also been suggested as a natural growth stimulant in different crops, helping in better fruit quality, improved seed production, and crop vigor [5].

### 3.4. Other Implications

Besides their role in nutrition, stress tolerance, and agriculture productivity, BRs are now studied for wider environmental and economic impacts. One possible benefit includes the reduction in pesticides and fertilizer application. As BRs increase immunity and tolerance in plants, it reduces the need for chemical intervention; hence, agriculture is more sustainable [26].

Brassinolide (BL) + Fe/Zn nanoparticles have great potential for the biofortification of plants, increasing their nutritional value and functional food production. Studies show that using 50 ppm ZnO-NPs with 10⁻⁸ M EBL increases plant growth, shoot length (39–44%), root length (28–33%), and dry biomass (37–46%). Photosynthetic efficiency also increases, with a 36% increase in chlorophyll content and 45% in the net photosynthetic rate (PN) [27].

Also, key antioxidant enzymes—catalase (CAT), peroxidase (POX), and superoxide dismutase (SOD) show a significant activity increase (60–76%), and plants become more resilient to oxidative stress. This combination increases nutrient accumulation, fruit yield and quality, lycopene and β-carotene content increase by 35% and 36%. Such a synergistic approach is a sustainable way to biofortify crops with micronutrients like Fe and Zn for nutrient-rich functional foods [27].

Moreover, priming seeds with zinc (Zn) and epibrassinolide enhances the expression of metal transporters, for example, ZIP and IRT1, which consequently increases seedling vigor, nutrient absorption, and stress resilience. Plant growth and resistance are improved by this procedure, which expands the accumulation of Fe and Zn. Given the rise in micronutrient deficiencies, epibrassinolide/Zn seed priming has great promise for biofortification and sustainable agriculture [28].

These BRs also help with mitigation potentials as they improve crops’ carbon sequestration abilities by improving crops’ ability to accumulate biomass and adapt to stress conditions [10]. This finds some global agricultural implications mainly in regions prone to environmental degradation and extreme weather conditions.

## 4. Micronutrient Homeostasis in Plants: Important Micronutrients for Plant Growth, Involvement of Signaling Molecules in Nutrient Uptake, and Stress Response

Micronutrient homeostasis is the maintenance of balance and the tight regulation of trace elements in plants such as manganese (Mn), zinc (Zn), copper (Cu), iron (Fe), and Boron (B), which are considered crucial for the development, growth, and physiological processes including photosynthesis and enzymatic activity [4]. This homeostasis is necessary for plants as it provides a balance that ensures that nutrient absorption, transport, storage, and utilization are done efficiently and avoids toxicity. Particularly under stress conditions, like nutrient-deficient soils, salinity, or drought, micronutrient homeostasis proves to be extremely crucial for stress resilience [29,30]. A deficiency of micronutrients in plants manifests as impaired plant growth and reduced yield, whereas an excess of micronutrients can cause toxicity, leading to impaired metabolic functions and damage to cellular structures [31]. Thus, maintaining equilibrium is imperative for plants to ensure optimal productivity and growth by adapting and surviving in varying environmental conditions.

Micronutrients are engaged with numerous physiological and biochemical processes that are pivotal to the development and growth of plants. Iron (Fe) is fundamental for the production of chlorophyll and is associated with both photosynthesis and respiration’s electron transport chains [32]. It is a fundamental part of the enzymes that produce energy and is needed to keep the chloroplasts functioning appropriately. Chlorosis, a condition where leaves become yellow from an absence of chlorophyll, can be brought by an iron deficiency and hinder plant development [33]. On the other hand, an excess of iron can result in reactive oxygen species (ROS), which damage cells [34].

Another crucial micronutrient is zinc (Zn), which is needed for protein synthesis, hormonal control (particularly auxin production), and enzyme activation [35]. Zinc affects the maturity of seeds and stalks and contributes to the production of chlorophyll [36]. Leaf chlorosis, necrotic spots, shorter internodal distance, stunted development, and malformed leaves can all be indications of zinc deficiency [37]. On the contrary, an excess of zinc can prevent the absorption of other essential elements, prompting nutritional imbalances in plants.

Manganese (Mn) in particular helps with splitting water molecules in the oxygen-evolving complex (OEC) of photosystem II, which is a critical part of photosynthesis [38,39]. A few other metabolic enzymes, including those that catalyze glycosylation and the scavenging of reactive oxygen species (ROS), likewise require it as a cofactor [38]. Manganese toxicity can cause leaf spotting and stunted growth attributable to enzyme inhibition, while manganese deficiency often leads to poor photosynthesis and decreased plant vigor [40].

In addition to being essential for electron transport as a Cytochrome C oxidase complex of the respiratory chain and redox processes, copper (Cu) additionally helps in lignin formation, which strengthens cell walls [41]. Furthermore, copper is fundamental for the activities of enzymes in photosynthesis and respiration [42]. Poor plant growth, wilting, and delayed flowering can be caused by a copper deficit, while excess copper can interfere with root development and nutrient uptake [43].

For reproductive development, membrane integrity, and cell wall construction, boron (B) is fundamental [43,44]. It promotes the production of seeds, cell division, and ion transport across membranes [43,44]. Poor root and flower development results from boron deficiency, while excess boron can impair photosynthetic capacity and prevent pollen germination, which brings down total plant yield [45]. For plants to thrive and produce at their best, micronutrient levels should be appropriately regulated.

The uptake and distribution of these micronutrients in plants are facilitated by various signaling molecules, particularly during biotic and abiotic stress conditions including metal toxicity, salinity, and drought. Boron is the only element that is taken up as an uncharged molecule instead of an ion in plants [46]. The primary regulation of boron (B) occurs through influx and efflux transporters. In the studies on Arabidopsis thaliana, it was found that the initial Boron uptake is facilitated by the transcriptional upregulation of transporter NIP5 under B deficiency [45,46]. In xylem loading and boron toxicity, a key role is played by the B exporter BOR1, while the distribution of B is facilitated by NIP6 [46].

For iron (Fe), a range of transporters (Table 2) such as OsIRT1, OsIRT2, IRT1, and transcription factors like the basic helix–loop–helix (bHLH) transcription factor AtFIT, IDEF1, and IDEF2 have been identified [47,48,49]. Zinc uptake and transport occur through various transporters (Table 2) such as ZIP (ZRT/IRT), HMA, VIT, NRAMP, and MTP [48]. Manganese transport occurs through transporters such as OsNRAMP5, OsMTP9, NRAMP, YSL, and ZRT/IRT [37]. For copper (Cu), the transporters involved are ZRT/IRT, YSL, HMA, and CDF [41]. During abiotic stress conditions, reactive oxygen species (ROS) are also considered to be crucial signaling molecules [50,51].

In the adaptive process of plants during stress, hormones, including ethylene, abscisic acid (ABA), and brassinosteroids (BRs) are fundamental [63]. For instance, ABA regulates root growth and stomatal closure, which indirectly influences the efficiency of nutrient intake and assists plants with managing drought stress [64,65]. In doing so, the plant can save its vital micronutrient supply while conserving water [64,65]. Like how ethylene and BRs modify root architecture and transport mechanisms to suit stressed plants’ demands, they also impact nutrient uptake [66]. Moreover, to support the maintenance of micronutrient homeostasis and enable the plant to endure and develop under unfavorable environmental circumstances, stress-responsive genes are triggered. A summarized mechanism is illustrated in Figure 3.

## 5. Role of Brassinosteroids in Iron (Fe) Homeostasis

For plants to carry out functions like respiration, photosynthesis, and DNA synthesis, iron (Fe) is a significant nutrient. Fe is abundant in the environment; however, its accessibility is often limited, especially in alkaline soils [70]. This can cause deficiencies that negatively affect plant development and nutritional quality. To maintain iron homeostasis, which is fundamental for growth, plants utilize various methods to control the intake and storage of iron. Two main Fe uptake strategies are utilized by plants. Strategy I involves reduction, essentially in non-grass plants, in which Fe^3+^ is reduced to Fe^2+^ for absorption. Conversely, grasses use strategy II, which relies upon phytosiderophores to chelate Fe^3+^, allowing greater absorption in alkaline conditions [70]. Excessive iron can cause oxidative stress, which disturbs physiological functioning and impairs nutrient absorption. Understanding Fe homeostasis is thus vital for increasing crop yields and nutritional content while likewise addressing global Fe deficiency issues in humans.

Brassinosteroids (BRs) are considered to be steroid hormones that are known to regulate various physiological processes of plants. Many studies have also focused on the role BRs play in plant stress responses, including mineral nutrient deficiency. The significant influence of BRs in iron (Fe) homeostasis in rice (*Oryza sativa*), particularly under Fe deficiency, was investigated, and it was found that the exogenous application of 24-epibrassinolide (EBR) increased leaf chlorosis and diminished growth in wild-type rice, showing greater vulnerability to Fe deficiency [71]. Conversely, the BR-deficient mutant d2-1 showed enhanced tolerance to Fe deficiency, with higher Fe accumulation in roots and the increased expression of Fe homeostasis-related genes like *OsIRT1* and *OsYSL15* under Fe-sufficient circumstances [71]. EBR increased the expression of genes associated with Fe absorption, which, however, repressed their expression in shoots, bringing down Fe transport and translocation. This shows that BRs assume a multifaceted role in modulating Fe homeostasis in rice by negatively regulating Fe uptake and transport under deficiency conditions, subsequently impacting plant development and nutritional status.

The role of 24-epibrassinolide (EBR) in advancing peanut seedling growth during iron deficiency was additionally analyzed. It was determined that EBR decreases the negative impacts of Fe deficit by enhancing chlorophyll content, soluble proteins, and proline while additionally improving antioxidant defenses against oxidative stress [72]. Previous studies demonstrate that H⁺-ATPase activates enzymes by acidifying the apoplast, which promotes root growth during nutrient deficiency. EBR likewise favorably affects the amounts of chlorophyll (Chl), a critical indicator of Fe status in plants [72]. Under Fe deficiency, chlorophyll production normally declines, causing decreased photosynthesis. Notwithstanding, EBR-treated plants had higher chlorophyll content, which was connected to the increased availability of Mg and active Fe, eventually boosting photosynthetic activity and plant development [72]. EBR application improved various physiological parameters, including ferric-chelate reductase (FCR) activity, antioxidant enzyme activity, and iron transport from roots to shoots. The ideal concentration for promoting growth was determined to be 10^−7^ M of EBR [72].

The investigation on the protective effects of 24-Epibrassinolide (EBR) on soybean plants under iron (Fe) deficiency revealed that EBR application expanded Fe uptake and accumulation in leaves, stems, and roots, bringing about improved nutritional content and metal homeostasis [73]. The investigation discovered that EBR treatment raised chlorophyll a (14%), chlorophyll b (23%), total chlorophyll (15%), and carotenoid levels (15%), enhancing photochemical efficiency and diminishing photoinhibition impacts in photosystem II [73]. Moreover, brassinosteroids (BRs), especially 100 nM of EBR, enhance iron (Fe) homeostasis in *Eucalyptus urophylla* through increasing Fe absorption and transport, employing increased H^+^-ATPase action in roots [74]. EBR likewise assists with lessening the Fe deficit by increasing the levels of other macronutrients and micronutrients, improving photosynthetic productivity, and lowering photoinhibition impacts on PSII. Mechanistically, EBR acts as a secondary messenger, activating antioxidant enzymes that reduce reactive oxygen species [74]. This action advances chlorophyll biosynthesis and improves plant growth by encouraging cell division and elongation, resulting in greater dry matter accumulation.

By inhibiting ferric reductase (FRO) activity and downregulating the expression of Fe transporters (CsFRO1 and CsIRT1), brassinosteroids (BRs) negatively control Fe-deficient responses in cucumbers [75]. In settings where there is sufficient Fe, BR administration increases FRO activity; in conditions where there is not sufficient Fe, it reduces Fe content in shoots and increases it in roots [75]. The response to Fe deficiency involves the formation of ethylene, and BRs demonstrate crosstalk with ethylene pathways. On the other hand, in rice plants exposed to Fe toxicity, brassinosteroids, particularly 24-epibrassinolide (EBR), are fundamental for maintaining iron (Fe) homeostasis. By enhancing the activity of significant enzymes like catalase and superoxide dismutase and lowering reactive oxygen species and oxidative damage, EBR treatment fortifies the antioxidant system [76]. EBR likewise decreases Fe buildup in roots and shoots and improves morphological features including nutrient absorption and root aerenchyma area. By promoting improved photosynthetic efficiency and overall plant development, this regulatory mechanism lessens the negative consequences of excess iron.

## 6. Role of Brassinosteroids in Zinc (Zn) Homeostasis

Zinc (Zn) is a plant micronutrient essential for several biological processes, such as enzymatic reactions, protein synthesis, and growth. Brassinosteroids (BRs) are a group of plant hormones that function as regulators in Zn homeostasis, mainly focusing on Zn uptake and the mobility and storage of this nutrient, destined to alleviate the symptoms related to its deficiency. Here, we discuss the roles of BRs in modulating plant Zn homeostasis, focusing on gene regulatory functions relevant to these processes [77,78].

Zinc uptake is controlled by the expression of the Zn transporter, mainly from the Zinc–Iron-regulated Transporter-like Protein (ZIP) family that is modeled by BRs. Crucially, these transporters are required for the efficient uptake of Zn from across the soil-root interface. It can further promote the transcription of these transporters and adapt Zn absorption in planta, such as under Zn-deficient conditions [79]. In addition, BRs appear to increase the root system by enhancing lateral roots that could provide a larger surface for Zn uptake in response to this hormone. Additionally, it leads to improved root architecture that facilitates the absorption of Zn by plants, which helps them perform their essential physiological processes such as enzyme activation and photosynthesis [80,81].

Besides modulating Zn uptake, BRs are essential in coordinating root-to-shoot zinc translocation. Studies have shown that BRs increase the biosynthesis of organic acids, citric and malic acid, among other soluble compounds capable of complex with Zn ions. It promotes the translocation of Zn through the xylem, which helps make the nutrient available to growing parts of the plant [78]. They are known to mobilize Zn from roots and, in turn, make it available for the efficient transport of Zn towards the shoot tissues that are essential for important metabolic processes, including photosynthetic energy. This skill to enhance the bioavailability of Zn is beneficial, especially when plants face scarcity of the nutrient, resulting in always meeting their nutritional and metabolic necessity [77].

Meanwhile, the crosstalk between BRs and plant hormones endorses Zn homeostasis at an even greater scale (Table 3). The regulation of root growth and hair formation by BRs often occurs through interactions with hormones, including auxins and gibberellins [78,79,80,81]. Not only does this cooperation improve Zn uptake efficiency, a more efficient distribution of Zn throughout the plant also usually results [61]. In addition, BRs inhibit ABA production, a hormone important to control the regulation of Zn uptake and plant stress management. This complex network of hormones is crucial for growth signals and provides a buffer against environmental stresses, such as in plants [43].

In the storage and distribution of Zn, BRs control cellular compartmentalization to store or distribute Zn in further subcellular parts. They activate the Zn-binding proteins, thus allowing them to concentrate zinc in compartments like vacuoles and other organelles. Such strategic sequestration protects plants from changes in Zn availability by maintaining an internal pool for essential metabolic processes [76]. Furthermore, BRs contribute to reducing symptoms of Zn deficiency by increasing antioxidant defenses. They induce the activity of enzymes such as superoxide dismutase (SOD) and catalase, which counteract oxidative damage frequently seen in cases of Zn deficiency. The BRs are able to improve the redox balance of plants and stabilize cellular structures, thus alleviating some negative effects resulting from Zn deficiency on growth/development [77,78].

In conclusion, the function of BRs in Zn homeostasis is thus nuanced. They control Zn uptake, enhance its mobilization, synergize with multiple hormones, balance storage, and alleviate deficiency symptoms. Taken together, these processes indicate the importance of BRs in the regulation of Zn nutritional homeostasis that is critical for plant health and productivity [70,74].

## 7. Role of Brassinosteroids in Manganese (Mn) Homeostasis

In plants, brassinosteroids (BRs) are essential for maintaining manganese (Mn) homeostasis and minimizing the negative effects of excess manganese and other ion stressors, such as salt. Even in the presence of manganese-induced stress, plants can continue to grow healthily because BRs preserve the balance of vital mineral elements. BRs accomplish this, for example, by controlling manganese absorption and distribution to maintain appropriate levels in plant tissues. Research demonstrates that manganese concentrations in the roots dramatically drop when BR levels are lowered, for instance by using propiconazole (PPZ), an inhibitor of BR production. This suggests that BRs play a crucial role in overall mineral homeostasis, stress tolerance, and sustaining appropriate manganese levels in roots [83].

Moreover, BRs promote manganese’s transfer from the roots to the shoots, facilitating its transport inside the plant. To maintain the right nutritional balance and promote overall plant growth and stress resilience, this distribution mechanism makes sure that the amounts of manganese in the shoots are optimum. Although BRs have no direct effect on the amount of manganese in the roots, their capacity to raise the concentration of manganese in the shoots emphasizes their significance in maintaining the proper distribution of this vital nutrient [71].

Furthermore, brassinosteroids stimulate the plant’s defense mechanism against oxidative stress driven by manganese poisoning. Reactive oxygen species (ROS) can be produced by manganese poisoning, which can cause cellular damage. Key antioxidant enzymes including catalase, peroxidase, and superoxide dismutase are activated more by BRs like 24-epibrassinolide (EBL), neutralizing ROS and reducing oxidative stress [70].

Additionally, BRs enhance the physiological characteristics and growth of plants subjected to manganese toxicity. They partially repair the harm produced by too much manganese by assisting in the restoration of water balance, photosynthetic efficiency, and overall plant growth. BRs play a broad function in enabling plants to grow and flourish in harsh conditions by assisting them in tolerating not just manganese stress but also other forms of heavy metal toxicity, such as cadmium and nickel. BRs help plants detoxify excess manganese, guaranteeing healthy development and enhanced stress tolerance by strengthening antioxidant defense mechanisms and encouraging appropriate nutrient distribution [84].

## 8. Role of Brassinosteroids in Copper (Cu) Homeostasis

Copper is an essential micronutrient that controls thousands of physiological processes in the plant, including photosynthesis and respiration as well as the synthesis of several critical biomolecules. Conversely, copper toxicity resulting from high concentrations leads to oxidative stress and cellular damage. Therefore, mechanisms must be developed to establish copper homeostasis. In these mechanisms, brassinosteroids, a family of plant hormones, are emerging as crucial regulatory components.

Brassinosteroids enhance tolerance in plants to copper-induced oxidative stress through optimization of the photosynthetic machinery and improvement in cellular viability. Experiments showed that BRs combined with hydrogen peroxide not only optimize stomatal movement and root morphology but also decrease the oxidative burst due to metal stress inside the roots of tomato plants [85]. This indicates that the role played by BRs under metal stress conditions is that of protectants because they enable cellular maintenance even under metal stress conditions. Brassinosteroids have further demonstrated their efficacy in aiding phytoremediation processes. They mediate oxidative damage from copper toxicity by increasing catalase (CAT), peroxidase (POX), and superoxide dismutase (SOD) activities. Additionally, they improve the root architecture with a stimulation of lateral root growth, thus allowing effective copper uptake and its segregation in less susceptible compartments with an overall reduction in cellular toxicity. This hormone also regulates the expression level of stress-responsive genes and transporters that will be inducing copper homeostasis and supporting plant tolerance. The integration of brassinolide into phytoremediation strategies renders an eco-friendly remedy to eradicate soils of copper contamination, thereby improving biomass production and sustainability in agriculture [84,86,87].

The 28-Homobrassinolide: Hydrogen peroxide (H_2_O_2_) plays a role as a mediator in the tolerance of *Vigna radiata* towards copper [84]. This interaction hints at the valuable role that such a pathway can play, suggesting that BRs can help the adaptive signaling pathways developed by a plant under copper stress. Brassinosteroids also affect root architecture, which improves nutrient uptake and stress tolerance. Elevated root morphology under heavy metal stress leads to increased water and nutrient intake, hence assisting in copper homeostasis [88]. The interaction of brassinosteroids with other phytohormones is key to copper homeostasis regulation. It further influences gene expression and metabolic pathways in contributions to copper tolerance [86].

Brassinosteroids regulate several physiological processes involved in enhancing plant tolerance towards copper stress. Mechanisms by which brassinosteroids may regulate such physiological events include the activation of antioxidant defense systems, alterations in the regulation of transporters that regulate ions, and the modulation of stress-responsive genes [86]. Such mechanisms are crucial to maintaining copper homeostasis and reducing the toxicity associated with copper.

Brassinosteroids provide a promising approach to enhance crop resistance against copper toxicity in contaminated soils. Apparently, the potential of BRs in agricultural practice hints that their application may improve crop yield and quality under stressful conditions [13]. In brief, brassinosteroids play a vital role in copper homeostasis as they will reduce oxidative stress, enhance root morphology, and interact with other hormones to heighten the resistance of plants. Their application would be one of the possible strategies used to increase crop resistance in copper-contaminated environments.

## 9. Role of Brassinosteroids in Boron (B) Homeostasis

Boron is a key micronutrient for multiple cellular activity in plants, such as the synthesis of cell walls, membrane function, and growth regulation. However, because of the unequal distribution of boron in soils, its deficit is a prevalent concern that has a big effect on vegetative well-being and crop vitality. A type of plant steroid hormone called brassinosteroids (BRs) is crucial for managing boron absorption, redistribution, and accumulation. BRs help plants respond to changes in boron availability [13,86].

BRs significantly influence boron absorption by altering the expression of transport-associated genes, such as BOR1 and NIPs. BR signaling upregulates the expression of these transporters under conditions of boron scarcity, enhancing optimal boron assimilation under constrained availability [89]. The modification of BR signaling pathways shows a correlation with increased boron absorption efficiency, which enables plants to sustain sufficient boron levels required for key cellular mechanisms involving cell wall integrity and growth processes [90].

BRs even enhance the boron distribution in all plant tissues, resulting in different transport-associated genes being expressed specifically at the meristematic regions to ensure that developing organs are properly supplied with boron [86]. This is crucial especially in the phase of relatively low levels of boron, when BRs aid in the transport of boron to the growing areas from older tissues [13]. This guided displacement of the boron is indicative of the internal mechanisms possessed by the plant to adapt to varying levels of boron in the soil and also demonstrates the importance of BRs in nutrient supply.

Uptake of boron and its distribution involve an interaction not only with BRs but also with other phytohormones such as auxins, gibberellins, and ethylene. They work on lateral roots with auxins to encourage stalks that pry further into the soil and enable an adequate sufficient area for the thirsted boron [91]. BRs also work together with C_2_H_4_ and psychrophilic reactive oxygen species to co-regulate stress-responsive genes that impact boron transporter activity, thereby enhancing plant resilience to changing external boron levels [92]. This work demonstrates the bifronted hormonal coordination of changes in the formative growth environment to stem nutritional insufficiency.

Boron is involved in the Extracellular Signal-Regulated Kinase (ERK)-mediated signaling pathways that regulate vacuolar transporters to set the level of boron storage and the spatial control of boron within the cell. BRs increase the activity of tonoplast transporters, which in turn allow the boron to be stored in vacuoles during the moments when the vacuoles are full, thus preventing the potential toxicity by eliminating the boron [93]. This regulated storage means that the plant can control the boron excess and reserve the material for the period of scarcity [13]. At the same time, BRs, unlike scavengers, move boron from the older to the developing parts during the growth period, which is required for the overall cellular wall function and, hence, reproductive success [94].

Boron starvation can be a major problem in the tissue’s plants; moreover, decreased growth, changes in the cell structure, and pectin synthesis are also found among other symptoms. Improved functioning of the antioxidant enzymes in the presence of BR treatment helps to offset the oxidative damage caused by boron deficiency [13].

Besides assisting in maintaining cellular stability in periods of limited boron supply, BRs further promote the synthesis of boron-binding proteins and vital cell wall components, such as pectin [94]. For vegetation to be robust and thrive even in conditions where soil boron concentrations are inadequate, various adaptive responses are crucial. Brassinosteroids hold a key part in managing boron levels, permitting plants to boost their intake, transport, preservation, and stress management approaches [90].

## 10. Future Prospect

The BRI1 receptor plays a major part in BR signaling, although little is known about its biological mechanisms, including whether it forms homo- or heterodimers, what role steroid-binding proteins play, and how autophosphorylation affects BR signaling. Gaining knowledge about these systems can help us control plant development more effectively. Through the use of sophisticated instruments such as mass spectrometry, scientists can better understand how BR signals are processed by examining the phosphorylation of BRI1 and associated proteins. Determining protein connections is essential for BR signaling. For instance, there may be connections between proteins such as TRIP-1 and BRI1, and current research is examining these interactions to learn more about the signaling pathway [12]. In the future, BRs present a multitude of opportunities:Increased Crop Yields: By promoting cell growth and division, BRs may raise crop output to fulfill the world’s food needs [18].Stress Tolerance: BRs are beneficial for crops in areas impacted by climate change because they help plants tolerate heat, salinity, and drought [61].Developments in BR Signaling: New proteins and pathways implicated in BR signaling will be found by the ongoing study, which will facilitate crop modification even further [95].Agricultural Applications: By lowering the need for pesticides and promoting growth in difficult environments, BRs could be commercialized for sustainable agriculture [93].Genetic Research: By identifying hundreds of BR-regulated genes, years of research on how these genes affect plant growth may be conducted, with the possibility of discovering new plant functions [86].New Research Areas: By enhancing plant responses like photosynthesis and nutrient efficiency and uncovering novel pathways, BRs may further aid in crop development [92].Model for Other Hormones: The understanding of BR signaling may be used in the research of other plant hormones, thereby expanding the field of plant biology [95].

In conclusion, BRs have a great deal of promise for raising crop yields, strengthening stress tolerance, and advancing sustainable agriculture. Further study is expected to reveal even more creative uses for BRs [12].

## 11. Conclusions

Brassinosteroids (BRs) are crucial plant hormones that influence plant development, growth, and responses to a range of biotic and abiotic stressors. This review focuses on their various capabilities in enhancing plant adaptability and nutrient management. BRs interact with other signaling pathways, for example, abscisic acid and auxins, assisting plants with better acclimating to environmental stresses including drought, salinity, and pathogen attacks. They regulate the absorption, distribution, and homeostasis of micronutrients like iron, zinc, manganese, copper, and boron, which are important for plant health. BRs optimize nutrient availability by the expression of transporter and stress-related genes, hence promoting plant metabolic activities and increasing stress tolerance. Moreover, BRs improve photosynthesis and chlorophyll production while fortifying antioxidant defenses and lowering oxidative damage under stressful situations. Since they boost nutrient uptake while simultaneously enhancing stress tolerance, BRs are a promising tool for sustainable agriculture. Better crop yields, higher-quality nutrition, and less dependence on chemical pesticides and fertilizers can all result from utilizing BRs. Future agricultural advancements will require BRs since their comprehension of BR-mediated pathways provides important insights for creating crop management plans that guarantee production even in subpar growing environments.

## Figures and Tables

**Figure 1 plants-14-00598-f001:**
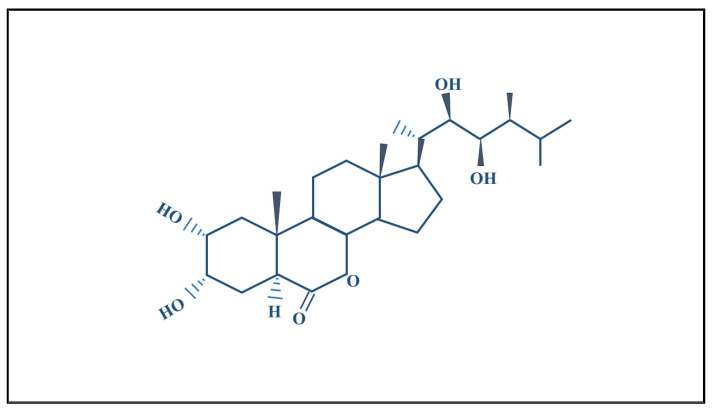
Chemical structural formula of brassinosteroids [12].

**Figure 2 plants-14-00598-f002:**
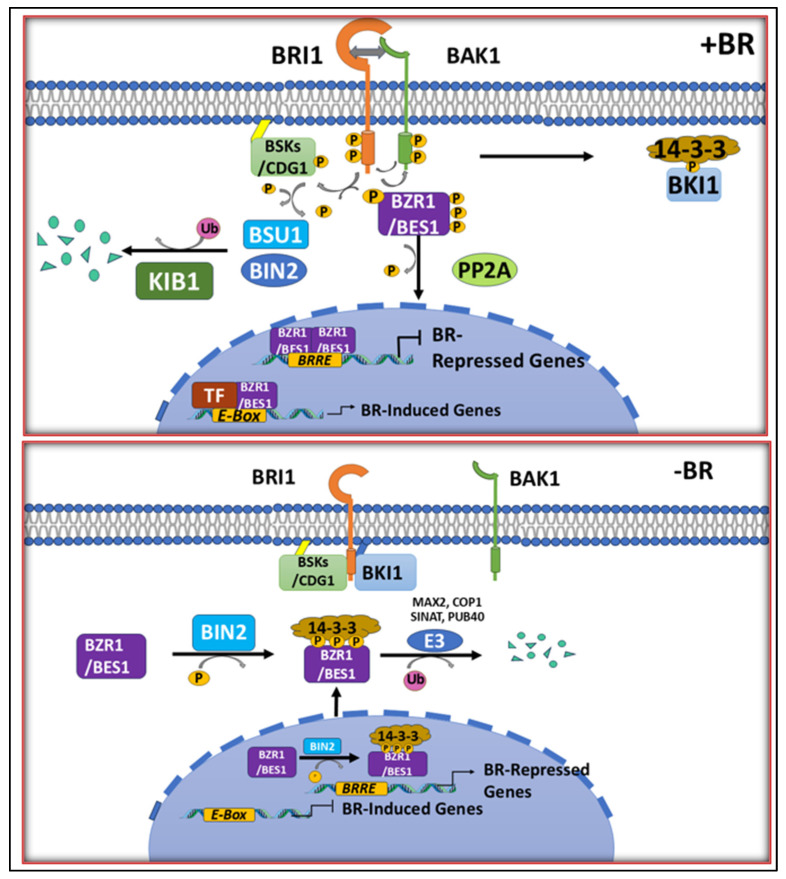
BR+ve: When BRs are available, BR binding to BRI1 and its co-receptor BAK1 sets off trans-phosphorylation between BRI1 and BAK1 and causes BKI1 to separate from BRI1. BSKs and CDG1 are downstream signaling parts that are phosphorylated by the active BRI1-BAK1 receptor complex. BR-ve: The interaction between BRI1 and BKI1 renders BRI1 inactive when brassinosteroids (BRs) are absent. The transcription factors BZR1 and BES1 get phosphorylated because of BIN2’s continued constitutive action because of this dormancy. By interacting with 14-3-3 proteins, phosphorylated BZR1 and BES1 are kept in the cytoplasm and stop their ability to bind DNA. Many E3 ligases, including MAX2, COP1, SINAT, and PUB40, help in their breakdown [16].

**Figure 3 plants-14-00598-f003:**
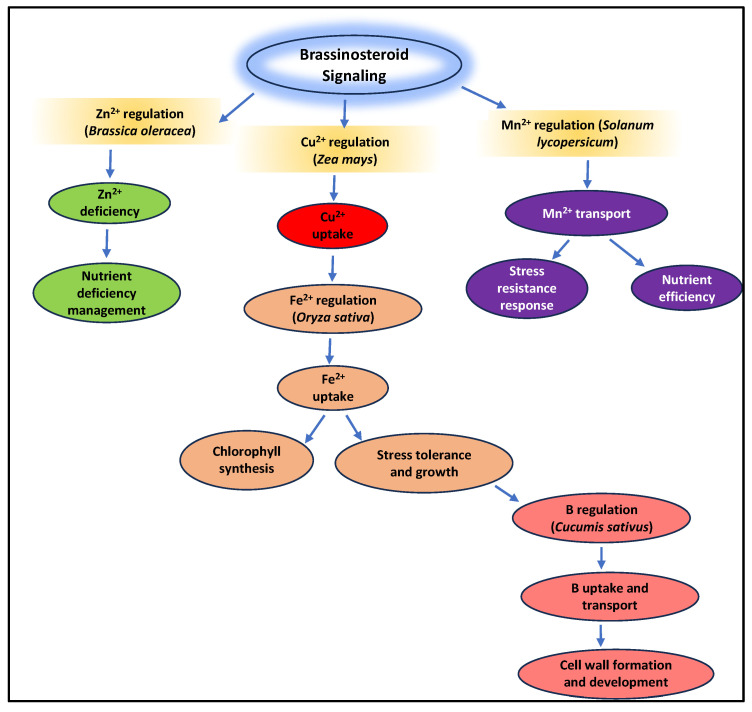
Brassinosteroid signaling and its role in micronutrient transport and plant development. This diagram illustrates the multifaceted roles of brassinosteroid signaling in regulating the uptake, transport, and utilization of key micronutrients in plants [9,43,67,68,69].

**Table 1 plants-14-00598-t001:** Structural description of brassinosteroids [12].

Feature	Description
**Chemical Nature**	Polyhydroxylated sterol derivative, structurally related to animal steroid hormones.
**Most Studied Brassinosteroid**	Brassinolide.
**Core Structure**	Steroidal lactone with a cholestane skeleton (5α-cholestan).
**A Ring**	Cis-vicinal hydroxyl groups at carbon positions C2 and C3.
**C Ring**	Six-membered lactone ring (C6–C7), distinguishing brassinosteroids from other steroids.
**C and D Rings**	Present in brassinosteroids.
**Side Chain**	Hydroxyl groups at C22 and C23; methyl group attached to C24.

**Table 2 plants-14-00598-t002:** List of transporters that aid in the homeostasis of micronutrients such as iron and zinc in plants and their specific role.

Transporters	Function	Micronutrient	Plant	References
IRT1	Transport of Fe^2+^ across the root cell membrane	Fe	Non-graminaceous strategy-I plants	[52,53]
YS1, YSL	Take up Fe(III)–Mas (mugineic acid family phytosiderophores)	Fe	Graminaceous strategy-I plants	[54]
OsIRT1	Its overexpression leads to Fe and Zn accumulation and it positively regulates saline–alkaline stress tolerance	Fe, Zn	*Oryza sativa* L.	[55,56,57]
FRD3, OsFRDL1	Helps in the citrate loading of xylem, which helps in Fe movement throughout the xylem	Fe	*Arabidopsis thaliana*	[55]
PEZ, PEZ2	Solubilizes apoplasmic iron contributing to long distance Fe transport	Fe	*Oryza sativa* L.	[58]
TOM1, TOM2, TOM3	Fe mobilization	Fe	*Oryza sativa* L.	[59]
OsVIT1, OsVIT2	OsVIT1 and OsVIT2 mutants cause increased Fe and Zn concentrations in rice seeds and decreased Fe and Zn concentrations in flag leaves	Fe, Zn	*Oryza sativa* L.	[60]
NRAMP3, NRAMP4	Transports Fe^2+^ and other divalent ions from vacuoles to cytoplasm	Fe	*Arabidopsis thaliana*	[61]
FRO2, FRO3, FRO7, FRO8	FRO2 reduces ferric iron chelates at the root surface–rhizosphere interface. FRO3 and FRO8 influences mitochondrial metal ion homeostasis. FRO7 contributes to Fe delivery to chloroplasts	Fe	*Arabidopsis thaliana*	[62]

**Table 3 plants-14-00598-t003:** Multifaceted roles of brassinosteroids (BRs) and Zinc (Zn) in enhancing plant biofortification, antioxidant defense, transporter regulation, and growth, highlighting key transporters.

Aspect	Role of BRs	Role of Zn	Transporter	Reference
Biofortification Potential	Enhances nutrient distribution to edible plant parts.	Improves zinc concentration in seeds and microgreens.	ZIP-mediated redistribution in seeds.	[70]
Antioxidant Defense	Stimulates antioxidant enzymes such as catalase and SOD.	Decreases the buildup of ROS during zinc deficiency or toxicity.	ROS regulation through catalase and SOD.	[82]
Transporter Regulation	Promotes gene expression for Zn transporters like ZIP and MTP families.	Facilitates the movement of zinc within tissues via MTPs (Metal Tolerance Proteins).	ZIP (ZIP5, ZIP7) and MTP (MTP1, MTP3).	[52]
Growth and Development	Boosts photosynthesis and biomass production.	Increases chlorophyll levels and supports protein synthesis.	Zn required for chlorophyll synthesis enzymes.	[5]

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
