# Peer review of "Brassinosteroids in Micronutrient Homeostasis: Mechanisms and Implications for Plant Nutrition and Stress Resilience"

_plants, 2025, doi:10.3390/plants14040598_

Round 1

Reviewer 1 Report

Comments and Suggestions for Authors

Dear authors. Due to the importance of epibrassinolides in plant physiology and agricultural practice  I suggest some revisions of the review which may improve the significance of the work:

1) explain why you have chosen Fe, Zn, Cu, Mn and B  for your review leaving aside two important immunomodulators: Se and Si

2) give a more broad description of brassinolide application in phytoremediation of areas polluted with copper. It seems not all literature data have been used.

3) provide additional data confirming high prospects of epibrassinolide supply for biofortification of plants with Fe and Zn for obtaining appropriate fuctional food. Include the data of Fe/Zn nanoparticles-epibrassinolide application on tomato growth and development (DOI: 10.32604/biocell.2021.015363 

3) add the description of epibrassinolide/Zn priming of seeds and indicate high prospects of such a technology.

Minor comments:

1) it is highly desirable to add a chemical structural formula of epibrassinolids as you try to describe it on lines 106-111

2) Table 1 Don't use predicate in the title

3) line 522 decipher ERK

4) Indicate that up-to-date more than 70 epibrassinolides have been identified and at present two: 28-homobrssinolide and 24-epibrassinolides are commercially available

Author Response

Reviewer 1 comments

Dear authors. Due to the importance of epibrassinolides in plant physiology and agricultural practice,  I suggest some revisions of the review which may improve the significance of the work:

Response: We thank the reviewer for the comments and suggestions. We have tried to answer all the points accordingly.

1) explain why you have chosen Fe, Zn, Cu, Mn and B  for your review leaving aside two important immunomodulators: Se and Si

Response: An explanation for why Fe, Zn, Cu, Mn and B have been selected and why Se and Si have been excluded has been provided in the introduction section for better understanding, as per the comment. selenium (Se) and silicon (Si), are considered beneficial immunomodulators involved in stress tolerance and plant defence mechanisms, but most (not all) are classified as non-essential or conditional micronutrients according to a plant species and environmental context. Additionally, the roles of BRs in the crosstalk with Se and Si are far less known compared to their established functions in regulating Fe, Zn, Cu, Mn and B, which have already been reviewed. This review, centred on these five micronutrients, aims to provide a targeted and in-depth analysis of their interactions with BRs, supported by robust experimental evidence.

2) give a more broad description of brassinolide application in phytoremediation of areas polluted with copper. It seems not all literature data have been used.

Response: In the Role of Brassinosteroid in Cu Homeostasis section, brassinolide application in phytoremediation of areas polluted with copper has been included with proper citations.

3) provide additional data confirming high prospects of epibrassinolide supply for biofortification of plants with Fe and Zn for obtaining appropriate fuctional food. Include the data of Fe/Zn nanoparticles-epibrassinolide application on tomato growth and development (DOI: 10.32604/biocell.2021.015363 

Response: As per the comments, the high prospects of epibrassinolide supply for biofortification of plants with Fe and Zn were incorporated in the article including the citation.

4) add the description of epibrassinolide/Zn priming of seeds and indicate high prospects of such a technology.

Response: The description of epibrassinolide/Zn priming of seeds and high prospects of such a technology has been included in the text.

Minor comments:

1) it is highly desirable to add a chemical structural formula of epibrassinolids as you try to describe it on lines 106-111

Response: The Chemical Structural Formula of Brassinosteroid has been added as Figure 1.

2) Table 1 Don't use predicate in the title

Response: Table 2’s heading has been adjusted.

3) line 522 decipher ERK

Response: ERK in the said lines has been better described.

4) Indicate that up-to-date more than 70 epibrassinolides have been identified and at present two: 28-homobrssinolide and 24-epibrassinolides are commercially available

Response: The provided information has been incorporated in the text.

Reviewer 2 Report

Comments and Suggestions for Authors

The manuscript, titled as ‘Brassinosteroids in Micronutrient Homeostasis: Mechanisms and Implications for Plant Nutrition and Stress Resilience’, showed us the important functions of brassinosteroids in plants for the uptake, distribution, and utilization of essential micronutrients through influencing the expression of genes responsible for the absorption and internal distribution of micronutrients. This manuscript can show the readers to understand the importance of BRs for nutrient use in plant productivity and sustainable agriculture. However, the manuscript needs further improvement.

Comments:

(1) Lines 104-111: I suggest, the features of this compound can be are presented in a table.

(2) The fonts in Figure 1 are too small and unclear. It is recommended to improve the quality of the figure.

(3) Section 2.4, the Signaling Pathways in Plant Growth and Development, I suggest that this section should describe the signaling transduction of BR in the uptake, distribution, and utilization of essential micronutrients. Or delete this section.

(4) Section 3, I think that this part of the content does not align closely with the theme of the manuscript. I suggest the author adjust the structure and framework of the manuscript. 

Author Response

Reviewer 2 comments

The manuscript, titled as ‘Brassinosteroids in Micronutrient Homeostasis: Mechanisms and Implications for Plant Nutrition and Stress Resilience’, showed us the important functions of brassinosteroids in plants for the uptake, distribution, and utilization of essential micronutrients through influencing the expression of genes responsible for the absorption and internal distribution of micronutrients. This manuscript can show the readers to understand the importance of BRs for nutrient use in plant productivity and sustainable agriculture. However, the manuscript needs further improvement.

Response: We thank the reviewer for the comments and suggestions. We have tried to answer all the points accordingly.

Comments:

(1) Lines 104-111: I suggest, the features of this compound can be are presented in a table.

Response: In addition to the diagram, the details of the Brassinosteroid have been given in the form of Table 1.

(2) The fonts in Figure 1 are too small and unclear. It is recommended to improve the quality of the figure.

Response: Figure 2 has been modified for better understanding.

(3) Section 2.4, the ‘Signaling Pathways in Plant Growth and Development’, I suggest that this section should describe the signaling transduction of BR in the uptake, distribution, and utilization of essential micronutrients. Or delete this section.

Response: Section 2.4 has been described in more details to incorporate the signaling transduction of BR in the uptake, distribution, and utilization of essential micronutrients. Deleting this section was not feasible as it indicates the signaling mechanism of BR in plants which is crucial in terms of the theme of the manuscript.

(4) Section 3, I think that this part of the content does not align closely with the theme of the manuscript. I suggest the author adjust the structure and framework of the manuscript. 

Response: Section 3: given the theme of the manuscript, which focuses on the role of Brassinosteroids in micronutrient homeostasis, this section seems to be relevant in gaining a broader understanding of what Brassinosteroid is and how it is relevant in a range of plant aspects, in addition to micronutrient homeostasis. That is why, we want to keep this section in the manuscript.

Reviewer 3 Report

Comments and Suggestions for Authors

The review article is based on evaluating the results of numerous studies on the importance of Brassinosteroids in plant nutrition and their resistance to environmental stresses. Microelements (iron, manganese, copper and boron) were analyzed. In the introduction, the authors presented the importance of the main microelements in plant nutrition. Then, they characterized the origin and importance of Brassinosteroids. The chapter is written correctly, but I propose adding a general structural formula of these compounds (in the form of a drawing) - line 111. In the next part of the work, the authors presented the role of Brassinosteroids in the homeostasis of the analyzed microelements. These chapters are written correctly based on the available literature. Knowledge of such mechanisms will allow for proper planning of plant fertilization in field crops in the future. In the final part of the work, the authors discussed the prospects for the use of Brassinosteroids in plant physiology and agricultural production.

The presentation of numerous studies (103 literature items) deserves positive attention. A reader interested in the problem reads the article with great interest. The article uses chapters and subchapters that facilitate the understanding of the presented content. The concise presentation of the problem in tables and figures also deserves positive attention. The article ends with a summary.

 After analyzing the article, I believe that in its current form, it can be published in the journal Plants

Author Response

Reviewer 3 comments

Comments and Suggestions for Authors

The review article is based on evaluating the results of numerous studies on the importance of Brassinosteroids in plant nutrition and their resistance to environmental stresses. Microelements (iron, manganese, copper and boron) were analyzed. In the introduction, the authors presented the importance of the main microelements in plant nutrition. Then, they characterized the origin and importance of Brassinosteroids. The chapter is written correctly, but I propose adding a general structural formula of these compounds (in the form of a drawing) - line 111. In the next part of the work, the authors presented the role of Brassinosteroids in the homeostasis of the analyzed microelements. These chapters are written correctly based on the available literature. Knowledge of such mechanisms will allow for proper planning of plant fertilization in field crops in the future. In the final part of the work, the authors discussed the prospects for the use of Brassinosteroids in plant physiology and agricultural production.

            The presentation of numerous studies (103 literature items) deserves positive attention. A reader interested in the problem reads the article with great interest. The article uses chapters and subchapters that facilitate the understanding of the presented content. The concise presentation of the problem in tables and figures also deserves positive attention. The article ends with a summary.

            After analyzing the article, I believe that in its current form, it can be published in the journal Plants

Response: We thank the reviewer for your supportive comments and we have also included the structure of the basic Brassinosteroid accordingly.

Round 2

Reviewer 2 Report

Comments and Suggestions for Authors

I have no other concerns.